# Combined Treatment with Ultrasound and Immune Checkpoint Inhibitors for Prostate Cancer

**DOI:** 10.3390/jcm11092448

**Published:** 2022-04-27

**Authors:** Fuuka Hayashi, Katsumi Shigemura, Koki Maeda, Aya Hiraoka, Noriaki Maeshige, Tooru Ooya, Shian-Ying Sung, Yong-Ming Yang, Masato Fujisawa

**Affiliations:** 1Department of International Health, Kobe University Graduate School of Health Sciences, Kobe 654-0142, Japan; fuuka.taylor@icloud.com; 2Department of Urology, Kobe University Graduate School of Medicine, Kobe 650-0017, Japan; kokimaeda1118@gmail.com (K.M.); yym1112@gmail.com (Y.-M.Y.); masato@med.kobe-u.ac.jp (M.F.); 3Department of Medical Technology, Kobe University of Medicine Faculty of Health Sciences, Kobe 654-0142, Japan; aya.kendobaka@gmail.com; 4Department of Rehabilitation Science, Kobe University Graduate School of Health Science, Kobe 654-0142, Japan; nmaeshige@pearl.kobe-u.ac.jp; 5John B. Little Center for Radiation Sciences, Harvard T.H. Chan School of Public Health, Boston, MA 02115, USA; 6Department of Chemical Science and Engineering, Graduate School of Engineering, Kobe University, Kobe 657-8501, Japan; ooya@tiger.kobe-u.ac.jp; 7Center for Advanced Medical Engineering Research & Development (CAMED), Kobe University, Kobe 650-0047, Japan; 8The Ph.D. Program for Translational Medicine, College of Medical Science and Technology, Taipei Medical University, Taipei 11031, Taiwan; ssung@tmu.edu.tw

**Keywords:** ultrasound irradiation, prostate cancer, apoptosis, immune checkpoint inhibitor

## Abstract

Background: Ultrasound (US) is mostly used for diagnostic purpose but could be used for cancer treatments with a US intensity or frequency fitted to such a purpose. Prostate cancer (PC) has the highest prevalence in the urological field, but indications for immune checkpoint inhibitors (ICIs) for PC are limited to very few cases. In this study, we compared the antitumor effect of US irradiation alone with the combined use of US and ICIs in vitro and in vivo. Methods: PC cell line TRAMP-C2 cells were used in our experiments. TRAMP-C2 cells were irradiated with US with pulse repeated frequencies (PRF) of 1, 10, and 100 Hz. Cell proliferation was evaluated by MTS assay and apoptotic cells were analyzed using flow cytometry. To verify the antitumor effect of US irradiation on PC in vivo, we conducted animal experiments using mice. TRAMP-C2-bearing mice were irradiated with US with PRF of 10 and 100 Hz. Three weeks after the start of US irradiation, anti-PD-1 antibody was administered to the mice. Finally, mice were sacrificed and tumors were collected. Immunohistochemical (IHC) analyses were assessed for cleaved caspase-3 and CD3 in tumor cell extracts. Results: Cell proliferation assays showed that 1 and 10 Hz US significantly inhibited cell survival (*p* < 0.0001). In addition, US irradiation induced apoptosis at 1, 10, and 100 Hz (*p* = 0.0129, *p* = 0.0150, and *p* = 0.0017, respectively). In animal experiments, a significant tumor growth inhibitory effect was observed at 10 and 100 Hz, and 100 Hz + ICIs (*p* < 0.05, respectively). Hematoxylin–eosin (H–E) staining showed a significant increase in the necrotic area of the tumor at 100 Hz and 100 Hz + ICIs (*p* < 0.05, respectively). In addition, under IHC staining the expression level of cleaved caspase-3 and the number of CD3-positive cells increased at 100 Hz (*p* < 0.05, respectively). Conclusion: US irradiation induced apoptosis in cells and reduced cell viability. In vivo tumor growth was suppressed by combined treatment with US irradiation and ICIs. Further research on immune system activation will lead to less invasive and more efficient treatments for PC.

## 1. Introduction

There are various treatment options for prostate cancer (PC), including active surveillance, surgery, radiation therapy, and hormone therapy, but complications such as erectile dysfunction and incontinence may occur with surgery, hematuria and bloody stools with radiation therapy, and the risk of progression to castration resistance with hormone therapy. Therefore, a less complicated and simpler treatment method is desired [1].

Immune checkpoint inhibitors (ICIs) are therapeutic agents that induce antitumor effects by activating autoimmunity. ICIs, such as targeting cytotoxic T lymphocyte antigen-4 (CTLA-4) and programmed death-1/ligand-1 (PD-1/PD-L1), have already been clinically applied to many cancers and are also used in the urological field for renal and bladder cancers [2,3,4]. However, the indications for ICIs are very limited for PC because of the extremely low number of immune cells such as T cells. Thus, the outcomes of ICI monotherapy are inadequate due to the poor immunological circumstance [5]. Recently, we found that US irradiation of PC cell lines can induce apoptosis without thermal damage to cancer cells [6]. If the antitumor effect of ICIs on PC is upregulated in the combination treatment with US irradiation, a less invasive and effective treatment of ICIs in clinical practice can be performed. 

Although US irradiation is mainly used for diagnostic purposes, we have reported that optimizing the intensity and pulse repetition frequency (PRF) of US revealed an antitumor effect on PC cell lines [6]. US is considered to be a frequency of 20 kHz or higher, but the waveform itself repeats on and off. The PRF is the frequency of its on-and-off cycles (Figure 1). By optimizing the PRF, an antitumor effect was observed without increasing the temperature of the irradiated area. We believe that the antitumor effect of US irradiation is due to cavitation, which may release tumor antigens in cancer cells by disrupting the cell membranes. 

We hypothesized that tumor immunity would respond to the release of cancer antigens due to the disruption of cell membranes by US irradiation and conducted experiments combining US irradiation with ICIs. In this study, we investigated the effects of combined treatment with US irradiation and ICIs in vitro and in vivo using a mouse prostate cancer cell line.

## 2. Materials and Methods

### 2.1. Cells

The transgenic adenocarcinoma mouse prostate (TRAMP)-C2 cell line can be used in studies to elucidate molecular mechanisms associated with the initiation progression and metastasis of prostate cancer. TRAMP is a transgenic line of C57BL/6 mice harboring a construct comprising the minimal −426/+28 rat probasin promoter (426 base pairs of the rat probasin gene promoter and 28 base pairs of 5′-untranslated region) driving prostate-specific epithelial expression of the SV40 large T antigen [7]. The TRAMP-C2 cell line was cultured in 35 mm dishes with Dulbecco’s Modified Eagle’s Medium (D-MEM) (High Glucose) with L-Glutamine and Phenol Red medium (Sigma-Aldrich Japan, Tokyo, Japan) supplemented with 10% fetal bovine serum (Sigma-Aldrich Japan) and 1% penicillin–streptomycin (Nacalai tesque, Kyoto, Japan). Cell cultures were maintained at 37 °C in a 5% CO₂ humidified atmosphere [8].

### 2.2. Ultrasound Irradiation Treatments

We irradiated TRAMP-C2 cells at different frequencies based on the theory that cavitation alone would damage cell membranes and induce apoptosis [6]. A lower repeated frequency of US irradiation causes a longer pulse per irradiation at the same irradiation time rate. US (3.0 W/cm^2^, 3 MHz, irradiation time rate: 20%) was applied to TRAMP-C2 cells for 5 min with a PRF of 1, 10, and 100 Hz using an ultrasound device (a modified model of the SZ-100, MINATO Medical Science, Osaka, Japan). In the animal experiments, we used only 10 and 100 Hz. The temperature was kept below 42.5 °C to avoid thermal necrosis.

### 2.3. Cell Viability

We conducted cell proliferation experiments to investigate changes in cell viability due to US irradiation of TRAMP-C2 cells. Cell proliferation was evaluated by MTS [3-(4,5-dimethylthiazol-2-yl)-5-(3carboxymethoxyphenyl)-2-(4-sulfophenyl)-22H-tetrazolium, inner salt] assay. Twenty-four hours after US irradiation, the cells were exposed to MTS for 4 h. Cell viability was determined by measuring absorbance at 490 nm using a spectrophotometer (Thermo Fisher Scientific, Tokyo, Japan). Each experiment was performed in triplicate. 

### 2.4. Apoptosis Assay

We conducted an apoptosis detection experiment to verify the apoptosis-inducing effect of US irradiation. TRAMP-C2 cells were seeded at 1.0 × 10⁶ cells/dish and cultured for 24 h at 37 °C and 5% CO₂. After culturing for 4 h or 24 h after US irradiation, apoptotic cells were analyzed by flow cytometry using the Annexin V-FITC Apoptosis Detection Kit (Nacalai tesque, Kyoto, Japan). Each experiment was performed in triplicate.

### 2.5. Animal Experiments

To verify the antitumor effects of US irradiation on prostate cancer in vivo, we conducted animal experiments using mice. All aspects of the experimental design and procedure were reviewed and approved by the institutional ethics and animal welfare committees of Kobe University. Five-week-old C57BL/6J mice were purchased from CLEA Japan, Inc. (Tokyo, Japan). One million TRAMP-C2 cells were inoculated at day 0 (*n* = 4, respectively). After the longest tumor diameter reached 7 mm, mice were randomly assigned to 5 treatment groups (10 Hz, 100 Hz, anti-PD-1, anti-PD-1 + 10 Hz, anti-PD-1 + 100 Hz) or a control group. US irradiation was performed once every two days, and tumor diameter measurement was performed once every three days. Tumor volumes were expressed by the following formula: (longest diameter) × (shortest diameter)^2^ × 0.5. In addition, 3 weeks after the start of US irradiation, anti-PD-1 antibody was administered once every 4 days for a total of 5 times. Anti-PD-1 antibody was administered intraperitoneally at 250 μg/dose. Finally, mice were sacrificed at day 82 and tumors were collected. Tumors were fixed and embedded with paraffin.

### 2.6. Immunohistochemical Staining

Paraffin-embedded tissue sections were deparaffinized and rehydrated. Antigen retrieval was performed in citrate buffer (pH 6.0) at 120 °C for 5 min. Immunohistochemical (IHC) staining was performed in an automatic tissue processor (Bond-Max, Leica Microsystems, Wetzlar, Germany) following the standard protocol. Briefly, tissue sections were incubated for 60 min with anti-cleaved Caspase-3, or anti-CD3 primary antibodies. Cleaved caspase-3 was used because its immunostaining method has a stable staining technic, good detecting sensitivity and excellent reproducibility [9]. After washing, sections were exposed to dextran polymer backbone-conjugated secondary antibodies with horseradish peroxidase for 12 min, according to the instrument’s standard protocols. Tissue sections were incubated with diaminobenzidine for 10 min and counterstained with hematoxylin. H–E staining was also performed to assess the necrotic area of the tumor. The resulting tissue slides were observed under a microscope BZ-X700 (Keyence, Osaka, Japan). 

### 2.7. Immunohistochemical Analysis

The stained slides were observed, and each was evaluated. For IHC staining using an anti-cleaved Caspase-3 antibody, the intensity of staining was evaluated on a 4-point scale of 0 (negative), 1 (weak), 2 (intermediate), and 3 (strong), and the range was evaluated on a three-point scale of 1 (0–10%), 2 (11–50%), and 3 (50% or more). The total IHC score was determined by multiplying the frequency and intensity scores [10]. In addition, for IHC staining using anti-CD3 antibody, we evaluated the ratio of the number of positive cells to the number of cells in the visual field. For H–E staining, the ratio of necrotic areas to the entire tumor was calculated.

### 2.8. Statistical Analysis

Comparisons between two different groups were performed using Student’s T-test. Comparisons between multiple groups were performed using one-way analysis of variance (ANOVA) followed by the Tukey–Kramer method. Statistical differences among means were considered significant when *p* < 0.05.

## 3. Results

### 3.1. Cell Proliferation Assay

We conducted cell proliferation experiments to investigate the cell proliferation inhibitory effect of US irradiation on TRAMP-C2 cells. Cell proliferation assays showed that US irradiation significantly suppressed cell growth by 65% at 1 Hz (*p* < 0.0001) and 10 Hz (*p* < 0.0001). US irradiation at 100 Hz showed a cell proliferation inhibitory effect of about 50%, but no statistically significant difference was obtained (Figure 2). 

### 3.2. Apoptosis Assay

Apoptosis detection experiments were conducted to verify that the cell proliferation inhibitory effect of US irradiation was caused by the induction of apoptosis. Four hours after US irradiation, early phase apoptosis was significantly induced at 1, 10, and 100 Hz compared to controls (*p* = 0.0129, *p* = 0.0150, and *p* = 0.0017, respectively), and late phase apoptosis was significantly induced at 1 and 10 Hz compared to controls (*p* = 0.0203 and *p* = 0.0468, respectively) (Figure 3A). Twenty-four hours after US irradiation, early phase apoptosis was significantly induced at 10 Hz (*p* = 0.0308), and late phase apoptosis was significantly induced at 1 and 10 Hz (*p* = 0.0277 and *p* = 0.0055, respectively) compared to controls (Figure 3B).

### 3.3. Animal Experiments

The therapeutic effects of US irradiation and ICIs were investigated in vivo using TRAMP-C2-bearing mice. No skin damage due to US irradiation was observed. In the animal experiments, on the 41st day after the start of irradiation a tumor growth inhibitory effect of about 75% was observed with 10 and 100 Hz of US irradiation and the combined use of US irradiation at 100 Hz and anti-PD-1 antibody compared to controls (Figure 4).

### 3.4. Immunohistochemical Analysis

Mouse tumors were examined for the in vivo inhibition of US and ICIs. The caspase-3 final score range were 2.45 ± 0.13, 2.45 ± 0.32, 3.7 ± 0.24, 1.00 ± 0.34, 1.6 ± 0.18, and 0.6 ± 0.21 (ctrl, 10 Hz, 100 Hz, ICIs, ICIs + 10 Hz, and ICIs + 100 Hz, respectively); the percentage of CD3 positive cells were 1.85 ± 0.54%, 7.12 ± 2.87%, 10.44 ± 1.11%, 6.11 ± 1.67%, 7.83 ± 2.97%, and 7.08 ± 2.16% (ctrl, 10 Hz, 100 Hz, ICIs, ICIs + 10 Hz, and ICIs + 100 Hz, respectively). Assessed by IHC staining, the expression level of cleaved Caspase-3 was about 1.5 times higher at 100 Hz than that of the controls (*p* = 0.0294) (Figure 5). In other words, US irradiation significantly induced apoptosis even in vivo. In addition, the proportion of CD3-positive cells significantly increased at 100 Hz by about 5.7 times compared to controls (*p* = 0.0349) (Figure 5). Assessed by H–E staining, the necrotic area of the tumor increased about 3.2 times at 100 Hz compared to controls (*p* = 0.0141) (Figure 6). The necrotic part of the tumor increased significantly with combined treatment by anti-PD-1 antibody and US irradiation (10 and 100 Hz) as compared with anti-PD-1 antibody monotherapy (*p* = 0.0339 and *p* = 0.0224) (Figure 6). Furthermore, when the overall image of the H–E-stained tissue and the IHC stained tissue with cleaved Caspase-3 were compared, the necrotic areas and cleaved Caspase-3 positive areas were almost the same (Figure 7). 

## 4. Discussion

There are various treatment options for prostate cancer, including surgery, radiotherapy, hormone therapy, and anticancer drugs as mentioned above, but immune checkpoint inhibitors are not currently playing a leading role. Reasons for this include extremely little intratumoral infiltration of cytotoxic T cells that attack cancer cells in PC tissue, and the presence of regulatory T cells that suppress tumor immunity around cancer cells. PC is currently considered an immunologically “cold” tumor [11]. The key to improving the response is considered to be the acquisition of an immune response via cancer antigen-specific T cells.

We reported that US irradiation induces apoptosis in PC cells by optimizing the PRF [6]. US is widely used in clinical practice as a diagnostic modality, but its use for therapeutic purposes is very limited, though it does include high-intensity focused ultrasound (HIFU). The therapeutic effect of US irradiation is thought to have two main mechanisms, the thermal effect and the cavitation effect [12]. The cavitation effect is caused by the formation of numerous tiny bubbles that burst when exposed to US. These shock waves are thought to cause cell apoptosis by making minute holes in the cell membrane [12].

In a study of high-intensity US irradiation and cancer immune response, combined HIFU and ICIs treatment for neuroblastoma, where ICI response is difficult, had an antitumor effect that could not be obtained by either treatment alone [13]. However, HIFU has thermal effects due to the aggregation of high-intensity US waves [14], raising concerns that it may cause tissue damage. Especially in PC treatment, it is essential to prevent complications from thermal damage to surrounding organs such as the urethra and rectum.

The advantage of our US irradiation over HIFU is that it irradiates the tissue without aggregating the US wave output from the conductor, producing antitumor effects without causing thermal tissue damage as seen with HIFU. Therefore, when considering future clinical applications, it may be possible to change PC cells immunologically from “cold” to “hot” while safely irradiating the entire prostate with dual therapeutic advantages for patients.

Although there have been several reports of US irradiation for lymphoma cells [15] and hepatocellular carcinoma [16], most were performed only in vitro. There are no reports of in vivo treatments in combination with immune checkpoint inhibitors, as far as we could find, nor did we find any reviewed reports of HIFU for prostate cancer or reports of side effects. Sehmbi et al. reviewed the literature on HIFU in the hepatobiliary system and reported about 15% skin burns as adverse events [17]. However, the temperature of the irradiated area was limited to 42.5 °C in the number of mice in our experiments.

In this study, we investigated whether optimizing the repetition frequency of US would have antitumor effects in vitro and in vivo on TRAMP-C2 cells. We further found that ICIs, which has been ineffective against PC, showed antitumor effects when used in combination with US irradiation.

In vitro, low frequencies (1 and 10 Hz) showed higher antitumor effects than high frequencies (100 Hz). This is consistent with the results previously published by Maeshige et al. [6]. In Zhou’s report, US irradiation with a lower PRF than 100 Hz caused strong cavitation and erosion at the surface of soft material [18]. The greater effect at 10 than 100 Hz can thus be explained by the induction of cavitation.

In vivo, the tumor volume results showed a significant tumor growth inhibitory effect in the 10 and 100 Hz US irradiation groups and in the combined ICIs and US irradiation (100 Hz) group compared with the control group. Unfortunately, no significant difference in tumor growth inhibitory effect was observed between the ICIs single agent group and the US + ICIs combination group. From the results of IHC staining performed using the removed tumor, the expression level of cleaved Caspase-3 increased in the group using only US irradiation treatment. However, in the groups using anti-PD-1 antibody, the expression level of cleaved Caspase-3 was not increased. Furthermore, the results of H–E staining of the removed tumor showed that the necrotic part of the tumor was significantly increased in the US + ICIs combination groups compared to the ICIs single agent group. Based on these results and in vitro results, it appears that US irradiation induces apoptosis in PC cells leading to cell death. The results show that ICIs monotherapy is not effective for PC, as in previous studies [5]. US + ICIs treatment induced necrosis in PC cells when the anti-PD-1 antibody, which is an ICI, was used. Previous reports have shown that ICIs alone has no therapeutic effect on PC [5]. However, since it was effective for tumor necrosis when used in combination with US in our study, ICI may find a new adjunct treatment role for PC.

Based on these results, we plan to further evaluate the immune system for tumor cell destruction by US irradiation. In addition, we may undertake the development of a balloon-type probe as a transrectal irradiation method.

We would like to emphasize the study limitations here. First, we used only TRAMP-C2 cell line: Our previous studies reported in vitro antitumor effects of US on human-derived PC cell lines, and androgen-independent (PC-3) and androgen-dependent (LNCaP) cell lines. In both cases, similar antitumor effects were observed [6]. Considering this report, the effect of US should be reproducible. Second, we only conducted the experiments using cancer cells in this study; thus, comparative experiments with normal prostate cells are needed in our future work. Third, the in vivo experiments used mice, and the experimental numbers were small. Fourth, we used only one type of ICIs in this situation, so we will conduct the experiments in combination with other types of ICIs such as PD-1 and CTLA-4 (see Introduction), or in combination with US and two drugs in our future work. Fifth, the ICI was intratumorally administered at the spot where the tumor was subcutaneously transplanted into the mice. Additionally, for the in vivo experiments, PRF was performed only at 10 and 100 Hz due to concerns about skin damage in mice, but we did not have any.

## 5. Conclusions

We found that by optimizing the US frequency, apoptosis can be induced in prostate cancer cells and cell viability can be reduced with safe and reliable results in vivo. In addition, the combination of US and ICIs was found to cause necrosis in tumor cells and suppress tumor growth. The results of this study suggest that ICIs may be a new adjunct treatment for PC when used in combination with US by converting immunologically “cold” PC to “hot” using US and ICIs. Future investigations are needed to expand the range of treatment conditions and explore the mechanisms of action behind these antitumor effects. Moreover, previous studies on PC reported that the phosphodiesterase type 5 inhibitor tadalafil showed a cell proliferation inhibitory effect on LNCaP cell lines in vitro. We are also considering a combination treatment of US and tadalafil, and expect further therapeutic effects [19].

## Figures and Tables

**Figure 1 jcm-11-02448-f001:**
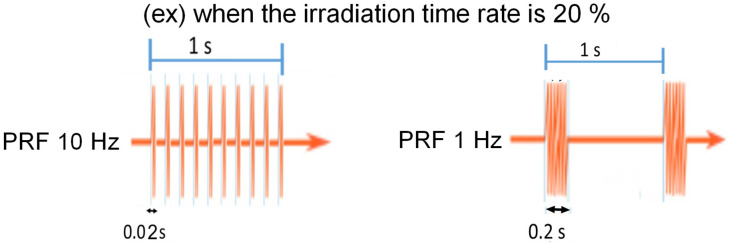
Ultrasound and pulse repetition frequency (PRF). PRF is the number of pulses of output per second in US waves. When the irradiation time rate is the same, the PRF is lower, and the longer the pulse per time.

**Figure 2 jcm-11-02448-f002:**
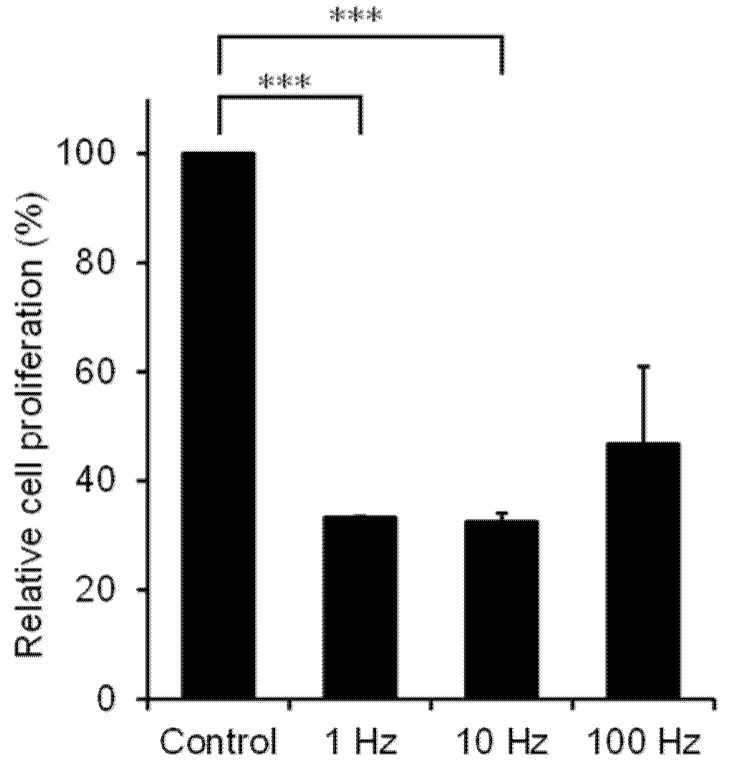
Cell proliferation inhibitory effect of US irradiation on TRAMP-C2 cells. TRAMP-C2 cells were irradiated with US, and cell proliferation was analyzed 24 h later by MTS assay (*n* = 3, average ± SE bars). Compared with controls, cell proliferation was significantly suppressed by US irradiation at 1 and 10 Hz (*p* < 0.0001, respectively) (***: *p* < 0.0001).

**Figure 3 jcm-11-02448-f003:**
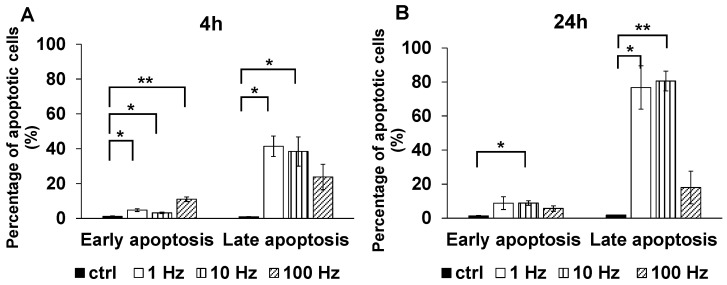
Detection of apoptotic cells 4 and 24 h after US irradiation. TRAMP-C2 cells were irradiated with US and after 4 h (**A**) or 24 h (**B**) an apoptosis detection experiment was performed using flow cytometry (*n* = 3). Both early and late phase apoptosis were significantly induced by US irradiation (*: *p* < 0.05, **: *p* < 0.01) except for late apoptosis at 100 Hz (4 h and 24 h) and for early apoptosis at 1 Hz and 100 Hz (24 h).

**Figure 4 jcm-11-02448-f004:**
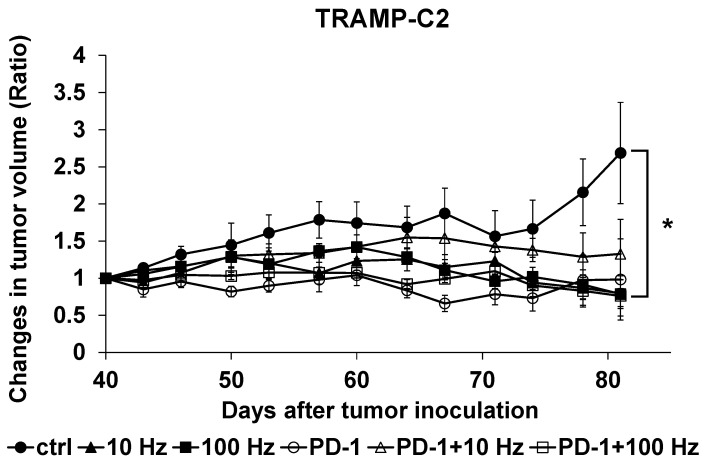
In vivo tumor inhibitory effects of US irradiation and ICIs. TRAMP-C2 cells were subcutaneously inoculated into C5 L/6J mice (*n* = 4). After tumor formation, mice were irradiated with ultrasound three times a week. At 14 days after the start of US irradiation, administration of anti-PD-1 antibody was started, and a total of 5 doses were administered. Significant therapeutic effects were observed with 10 and 100 Hz US irradiation, and in the combination treatment group with anti-PD-1 antibody and 100 Hz ultrasound irradiation (*: *p* < 0.05).

**Figure 5 jcm-11-02448-f005:**
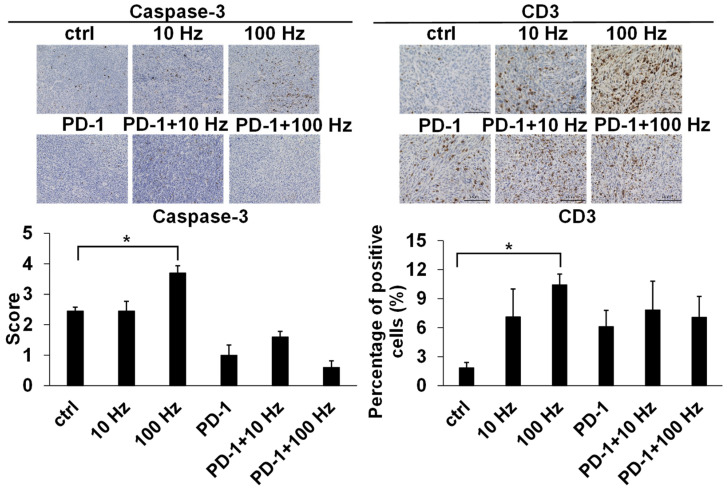
Immunohistochemical analysis of TRAMP-C2 mice tumors after US irradiation and ICIs treatment. After performing US irradiation treatment and US + ICIs combination treatment on TRAMP-C2 mice, immunohistochemical staining was performed on the removed tumor. IHC analysis showed that the expression level of cleaved Caspase-3 and the ratio of the number of CD3 positive cells were both significantly increased by US irradiation at 100 Hz (*: *p* < 0.05).

**Figure 6 jcm-11-02448-f006:**
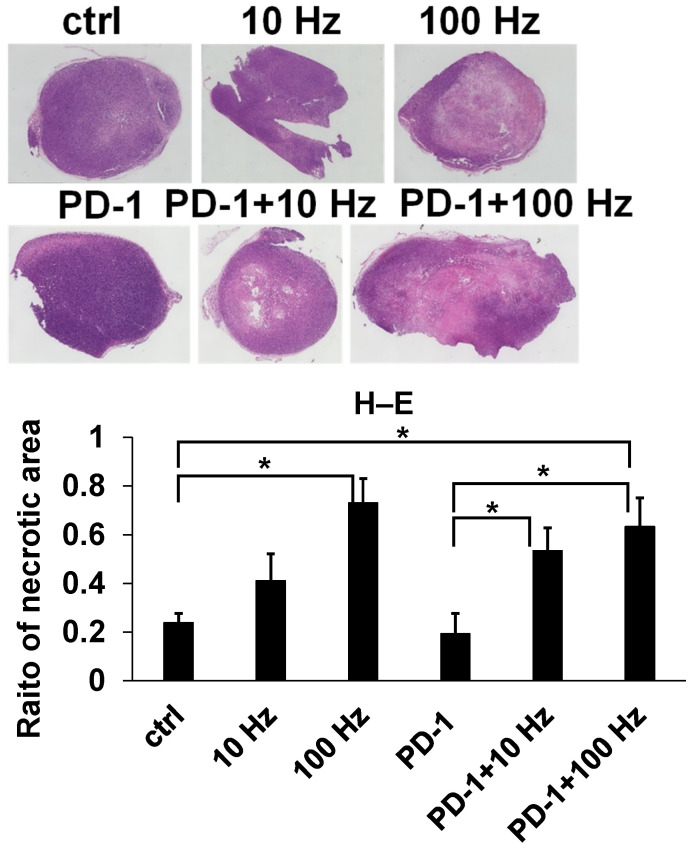
H–E stain analysis of TRAMP-C2 mice tumors. After US irradiation treatment and US + ICIs combination treatment in TRAMP-C2 mice, H–E staining was performed on the removed tumors. In the H–E staining analysis, the area of necrosis was measured with respect to the area of the entire tumor. The necrotic area is a pale pink (*: *p* < 0.05).

**Figure 7 jcm-11-02448-f007:**
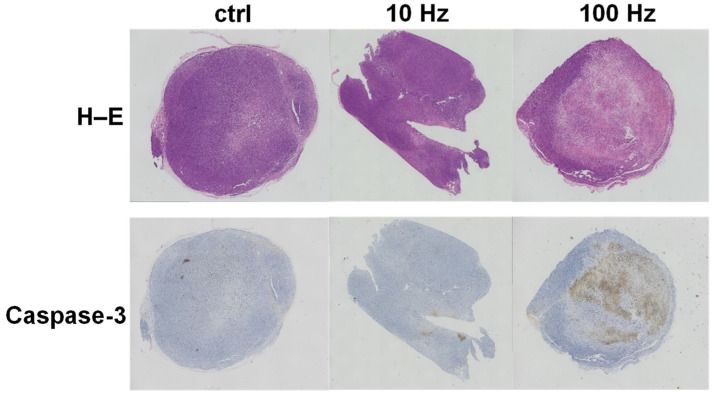
Comparison of H–E staining results and immunohistochemical staining results with anti-cleaved Caspase-3 antibody. We compared the results of H–E staining and immunohistochemical staining with anti-cleaved Caspase-3 antibody performed on post-treatment tumors removed from TRAMP-C2 mice. The photographs show the result of staining of the same individual. The necrotic part and the cleaved Caspase-3 positive part are almost the same, showing that apoptosis induction caused necrosis and suppressed tumor growth.

## Data Availability

The data are available from the corresponding authors upon reasonable request.

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
