# Peer review of "Combined Treatment with Ultrasound and Immune Checkpoint Inhibitors for Prostate Cancer"

_jcm, 2022, doi:10.3390/jcm11092448_

Round 1
Reviewer 1 Report
The manuscript entitled:" Combined treatment with ultrasound and immune checkpoint inhibitors for prostate cancer" focused on the evaluation of ICis plus US in the treatment of prostate cancer requires several major revisions to be accepted for the publication:
- In the study design section, the authors show that a single cell line was adopted to perform treatment evaluation. According to this point, the adoption of more than one cell lines may represent a crucial point to evaluate reproducibility of this approach.
- In the material and method section, the authors focus on prostate cancer cell line TRAMP- 18 C2. In my opinion, the adoption of a control group may represent a point of strenght for the statistical analysis. In addition, could the authors also elucidate the molecular profile of TRAMP- 18 C2? Could this line harbor a molecular assessment respopnsive to this approach?
- In the text, the authors also indicate ICs drugs in combination with US. In the study design section, the authors only focus on anti PD-L1. Could it represent a bias for the study?
- In the experimental section, the authors focus on IHC approach for the evaluation of cell proliferation. As regards, i would suggest to implement this section by performing other tests able to elucidate this condition
- In the introduction section, please, could the authors report how ICs drugs may be approached in the clinical managment of prostate cancer patients?
Author Response
Reviewer 1
<Comments and Suggestions for Authors>
The manuscript entitled:" Combined treatment with ultrasound and immune checkpoint inhibitors for prostate cancer" focused on the evaluation of ICis plus US in the treatment of prostate cancer requires several major revisions to be accepted for the publication:
In the study design section, the authors show that a single cell line was adopted to perform treatment evaluation. According to this point, the adoption of more than one cell lines may represent a crucial point to evaluate reproducibility of this approach.
(Amendments)
Thank you for the comments.
Our previous studies have reported in vitro antitumor effects of US on human derived PC cell lines, androgen-independent (PC-3) and androgen-dependent (LNCaP) cell lines, where the same anti-tumor effect was seen [6]. Considering this report, the effect of US should be reproductible. We have stated in the study limitation in discussion at line 321-325 on pg 10.
In the material and method section, the authors focus on prostate cancer cell line TRAMP- C2. In my opinion, the adoption of a control group may represent a point of strenght for the statistical analysis. In addition, could the authors also elucidate the molecular profile of TRAMP- C2? Could this line harbor a molecular assessment respopnsive to this approach?
(Amendments)
Thank you for the comments.
We will conduct comparative experiments using normal prostate cells in future research and have added this as the study limitation in Discussion at line 325-327 on pg 10.
In addition, TRAMP-C2 cell line can be used in studies to elucidate molecular mechanisms associated with the initiation progression and metastasis of prostate cancer This cell line is also a useful tool for gene drug discovery. TRAMP is a transgenic line of C57BL/6 mice harboring a construct comprised of the minimal -426/+28 rat probasin promoter (426 base pairs of the rat probasin gene promoter and 28 base pairs of 5'-untranslated region) driving prostate-specific epithelial expression of the SV40 large T antigen [7]. We have added these in Methods at line 82-87, on pg 2.
In the text, the authors also indicate ICs drugs in combination with US. In the study design section, the authors only focus on anti PD-L1. Could it represent a bias for the study?
(Amendments)
Thank you for the comments.
In this study, we used only one type of ICIs to see if it would work in combination with the US. We will conduct experiments in combination with cytotoxic T lymphocyte antigen-4 (CTLA-4) and programmed death-1 / ligand-1 (PD-1 / PD-L1), or in combination with US and two drugs in future work.
We have added this as the study limitation in Discussion at line 328-331 on pg 10.
In addition, we added a further description of ICIs that ICIs includes targeting cytotoxic T lymphocyte antigen-4 (CTLA-4) and programmed death-1 / ligand-1 (PD-1 / PD-L1) [2] to Introduction at line 52-55 on pg 2.
In the experimental section, the authors focus on IHC approach for the evaluation of cell proliferation. As regards, I would suggest to implement this section by performing other tests able to elucidate this condition
(Amendments)
Thank you for the comments.
Since we are conducting cell proliferation experiments in vitro and measuring tumor volume in vivo, we think that verification of cell proliferation is sufficient.
In the introduction section, please, could the authors report how ICs drugs may be approached in the clinical management of prostate cancer patients?
(Amendments)
Thank you for the comments.
Currently, we have found that US irradiation with even 1day of 10 Hz to PC cell lines can induce apoptosis without thermal damage to cancer cells [6]. From this perspective, if the effect of ICIs on PC is upregulated when used in combination with US irradiation, effective treatment with less burden in clinical practice can be performed.
We have added this explanation in Introduction at line 59-61 on pg 2.
Reviewer 2 Report
General comments:
The aim of this study was to examine anti-tumor effects of Ultrasound (US) irradiation alone and anti-tumor effects of combined use of US and immune checkpoint inhibitors (ICI) on prostate cancer (PC), in vitro and in vivo. The authors found that by optimizing the US frequency, apoptosis can be induced in prostate cancer cells and cell viability can be reduced in vivo. Moreover, they showed that the combination of US and ICI can cause necrosis in tumor cells and suppress tumor growth. The authors concluded that ICI may be a new adjunct treatment for PC when used in combination with US by converting immunologically “cold” PC to “hot”.
I find these results very interesting and potentially important for further investigations on prostate cancer (PC), possibly, other types of cancer. Importantly, the authors stated all limitations of their approach and I have nothing to add. In spite of that I think that the paper is worth publishing. Overall, the manuscript is clear and flows well but there are several points that need to be taken into consideration:
Specific comments:
IHC analysis and scoring is not clear and need to be clarified. First, which antibodies were used, primarily for Caspase 3(not specified in Material and Method section)? Did the authors use antibodies for cleaved caspase-3 or caspase-3? And, why? Second, how did the authors choose the scoring method for IHC analysis? The intensity score and percentage of positive cells are usualy added to produce the final scores. What is the range of final scores?
Author Response
Reviewer 2
<Comments and Suggestions for Authors>
The aim of this study was to examine anti-tumor effects of Ultrasound (US) irradiation alone and anti-tumor effects of combined use of US and immune checkpoint inhibitors (ICI) on prostate cancer (PC), in vitro and in vivo. The authors found that by optimizing the US frequency, apoptosis can be induced in prostate cancer cells and cell viability can be reduced in vivo. Moreover, they showed that the combination of US and ICI can cause necrosis in tumor cells and suppress tumor growth. The authors concluded that ICI may be a new adjunct treatment for PC when used in combination with US by converting immunologically “cold” PC to “hot”.
I find these results very interesting and potentially important for further investigations on prostate cancer (PC), possibly, other types of cancer. Importantly, the authors stated all limitations of their approach and I have nothing to add. In spite of that I think that the paper is worth publishing. Overall, the manuscript is clear and flows well but there are several points that need to be taken into consideration:
Specific comments:
IHC analysis and scoring is not clear and need to be clarified. First, which antibodies were used, primarily for Caspase 3(not specified in Material and Method section)? Did the authors use antibodies for cleaved caspase-3 or caspase-3? And, why?
(Amendments)
Thank you for the comments.
We used cleaved caspase-3 because its immunostaining method has a stable staining technic, good detecting sensitivity and excellent reproducibility [9].
We added this to methods at line 139-141 on pg 4.
Second, how did the authors choose the scoring method for IHC analysis? The intensity score and percentage of positive cells are usualy added to produce the final scores. What is the range of final scores?
(Amendments)
IHC scoring was based on the percentage of positive cells. The staining intensity was scored as 0 (negative), 1+ (weak), 2+ (medium) or 3+ (strong). The percentage of stained cells was categorized as: 1, 0–10%; 2, 11–50%; and 3, more than 50% stained cells. The total IHC score was determined by multiplying the frequency and intensity scores [10]. We have added these things to methods at line 154 on pg 4.
Furthermore, the caspase-3 final score range were 2.45±0.13, 2.45±0.32, 3.7±0.24, 1.00±0.34, 1.6±0.18, 0.6±0.21 (ctrl, 10Hz, 100Hz, ICIs, ICIs+10Hz, ICIs+100Hz). And the percentage of CD3 positive cells were 1.85±0.54 %, 7.12±2.87 %, 10.44±1.11 %, 6.11±1.67 %, 7.83±2.97 %, 7.08±2.16 % (ctrl, 10Hz, 100Hz, ICIs, ICIs+10Hz, ICIs+100Hz). We have added these data to 3.4 immunohistochemical analysis in Results at line 212-216 on pg 6-7.
References
[2] Daniel, Y.W.; Joe-Elie, S. et al. Fatal Toxic Effects Associated With Immune Checkpoint Inhibitors. JAMA Oncol. 2018, 12,1721-1728. doi: 10.1001/jamaoncol.2018.3923.
[6] Noriaki M.; Koichi K. et al. Can ultrasound irradiation be a therapeutic option for prostate cancer? Prostate. 2020, 80, 986-992. doi: 10.1002/pros.24030.
[7] Foster B., Gingrich J. et al. Characterization of prostatic epithelial cell lines derived from transgenic adenocarcinoma of the mouse prostate (TRAMP) model. Cancer res. 1997, 16, 3325-30.
[9] Rachel W., Debra G. et al. Comparison of immunohistochemistry for activated caspase-3 and cleaved cytokeratin 18 with the TUNEL method for quantification of apoptosis in histological sections of PC-3 subcutaneous xenografts. Comparative study. 2003, 199, 221-8.
[10] Kitagawa K, et al. Possible correlation of sonic hedgehog signaling with epithelial–mesenchymal transition in muscle-invasive bladder cancer progression. J. Cancer Res. Clin. Oncol. 2019, 145, 2261–2271. doi: 10.1007/s00432-019-02987-z.

Round 2
Reviewer 1 Report
The authors have addressed all the comments.
Author Response
Thank you so much for your checking.